# Acute Pulmonary Histoplasmosis Outbreak in A Documentary Film Crew Travelling from Guatemala to Australia

**DOI:** 10.3390/tropicalmed4010025

**Published:** 2019-02-01

**Authors:** Stephen Muhi, Amy Crowe, John Daffy

**Affiliations:** St Vincent’s Pathology, St Vincent’s Hospital (Melbourne), Victoria 3065, Australia; Amy.Crowe@svha.org.au (A.C.); john.daffy@svha.org.au (J.D.)

**Keywords:** histoplasmosis, travel, outbreak, Guatemala

## Abstract

*Histoplasma capsulatum* is an endemic mycosis with a widespread distribution, although it is infrequently reported in travellers. In April 2018, five television crew members developed an acute febrile illness after filming a documentary about vampire bats in Guatemala. Patients developed symptoms after travelling to Australia, where they presented for medical care.

## 1. Case Report

From 24 March to 4 April 2018, 12 television crew members filmed a nature documentary about vampire bats in the Cueva de Juan Flores, Petén Department, Guatemala. All film crew members received general pre-travel assessment and counselling, including recommended pre-travel vaccinations, including influenza, hepatitis A, typhoid and rabies vaccination. Although crew members were encouraged to wear a mask as a general safety measure, they were periodically removed due to perceived discomfort in the humid conditions of the cave.

The television crew then travelled to Australia, where some members flew to Melbourne to begin filming another nature documentary, while the others remained in Sydney. Two patients presented to our health service (both camera operators), while the other three presented to local health services in Sydney, Australia (an audio engineer, television producer and host). The two patients who presented to our health service were both males, aged 33 (patient 1) and 54 (patient 2), with no significant past medical history. A written informed consent for publication was obtained from both patients. Their symptoms presented over the course of 24 h, including severe fatigue, neck and shoulder pain, headache, fevers, chills and cough. Symptom onset in both patients was 8 days after completion of filming.

On examination, both patients were febrile between 39–41 °C and haemodynamically stable. Oxygen saturation was ≥ 97% in both patients. Physical examination of both patients was otherwise unrevealing, with a clear chest on auscultation and no lymphadenopathy or organomegaly. Full blood examination was within normal limits for both patients, with frequent atypical and reactive lymphocytes on blood film. Both patients had normal renal and hepatic function with moderately elevated C-reactive protein. Both patients also received a chest x-ray, which was unremarkable in patient 1. The chest x-ray of patient 2 showed mild patchy reticular markings, but was otherwise normal. Additional investigations for influenza, malaria, typhoid, leptospirosis, rickettsia, arboviruses and atypical pneumonia were negative.

Given the exposure of both patients to caves in Central America, histoplasmosis serology was performed via immunodiffusion (Westmead Hospital, Sydney, Australia) and a urinary antigen (by EIA) was sent to Indianapolis, Indiana (MiraVista Diagnostics). Patient 2 provided a sputum sample, which was sent to Westmead Hospital, Sydney for prolonged incubation fungal culture.

Both patients were monitored for 72 h, by which point symptoms had significantly improved without antifungal therapy, and treatment was supportive, in concordance with IDSA guidelines [1]. Subsequent correspondence with both patients confirmed recovery by 4 weeks. Clinical information regarding the three unwell film crew members who travelled to Sydney was unavailable to our unit.

Histoplasmosis serology was initially negative by immunodiffusion in both patients (Westmead Hospital, Sydney, Australia). Histoplasma urinary antigen (MiraVista, Indianapolis, IN, USA) was detected in both patients, with a turn-around time of 13 days. The diagnosis was subsequently confirmed serologically in patient 1, who returned to his primary healthcare physician in the United States one week after discharge, where complement fixation demonstrated a 1:16 titre to yeast-phase antibody, and immunodiffusion detected a positive M band (Quest Diagnostics, West Hills, CA, USA). In patient 2, the diagnosis was confirmed in sputum, which was culture-positive after 4 weeks.

## 2. Discussion

The key challenge for clinicians reviewing travellers from regions endemic for *H. capsulatum* is the non-specific presentation of patients with acute pulmonary histoplasmosis (APH), which may mimic a number of bacterial and viral infections seen in returned travellers. A unique form of epidemiological evidence available to clinicians in this case included video footage viewed by the treating team, which confirmed hundreds of bats flying overhead in a swarming fashion, probably disturbed by the human activity and exposure to artificial light. Bat guano was also recorded on film to be falling directly onto our patients within the caving system (Figure 1, recorded by patient 1).

An increasing number of clustered cases in travellers have been reported in the literature, largely due to a rise in international travel and increasing rates of ecotourism (Table 1) [2,3,4,5,6,7,8]. The generally high attack rate is indicative of the large inoculum of infection, as exposure to caves and bat guano remains a key feature in the majority of previous reports. Diagnostic testing and treatment in these published reports also varies widely. Despite endemnicity in Australia, the unavailability of histoplasmosis urinary antigen continues to impair rapid diagnostic testing, particularly in acute cases, prior to seroconversion and culture positivity. 

Our cases demonstrate the utility and feasibility of performing urinary antigen testing, which is reportedly the most sensitive test to diagnose APH [9]. Although APH is cross-reactive with other endemic mycoses, this is not a consideration in cases acquired in Australia, further supporting previous calls for its introduction in a national reference laboratory [10]. When used alone to diagnose APH, urinary antigen is limited by its poor overall performance [11]. However, a more recently developed EIA (MiraVista Diagnostics) measures both immunoglobulin G (IgG) and IgM, reporting a sensitivity of 96.3% for APH, when combining both antigen and antibody assays [11]. As demonstrated in the case of patient 1, convalescent serology should be repeated after several weeks, if negative during initial testing.

The severity of illness in our patients was mild, requiring a short period of hospitalisation for observation, diagnostic work-up and supportive management. It is difficult to quantify the inoculum of exposure, as masks were intermittently removed during filming. Patients were educated on the need for compliance with personal-protective equipment, and also counselled regarding the risk of reactivation in the event of immunosuppression. Both of our patients travelled overseas soon after their hospitalisation. These cases continue to highlight the dynamic interaction between adventure tourism and human curiosity, trans-continental travel and the evolving human-animal-ecosystem interface.

## Figures and Tables

**Figure 1 tropicalmed-04-00025-f001:**
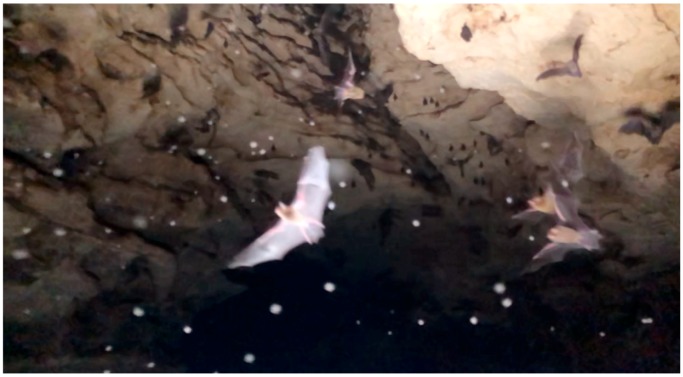
Image of bats filmed by patient 1 (particulate matter seen falling from above in foreground).

**Table 1 tropicalmed-04-00025-t001:** Outbreaks of acute pulmonary histoplasmosis in travellers from South and Central America.

Location	Year	Population/Activity	Attack Rate, %	Diagnosis	Treatment (%, Primary Indication)
Ecuador (2)	1999	US high school students cave exploring	11/17 (65%)	Urine Ag 0/2 (0%) Serology: 4/7 (57%)	3/17 (18%, prolonged symptoms)
Nicaragua (3)	2001	US “adventure travellers” cave exploring	12/14 (85%)	Urine Ag 7/12 (58%) Serology 14/14 (100%)	9/12 (75%, symptom severity)
Belize (4)	2002	Canadian high school students cave exploring	14/15 (93%)	Urine Ag 5/7 (71%) Serology: 3/15 (20%)	1/15 (7%, prolonged symptoms)
Guatemala, El Salvador (5)	2004	Norwegian tourists cave exploring	16/19 (84%)	Serology: 8/14 (57%)	3/16 (19%, not reported)
El Salvador (6)	2008	US missionaries renovating a church	20/33 (61%)	Antigen (serum/urine) 7/20 (35%)	Not reported
Ecuador (7)	2012	Polish tourists (organised tour) cave exploring	4/4 (100%)	Serology: 4/4 (100%)	2/4 (50%, prolonged symptoms)
Brazil (8)	2013	Scientists, researching histoplasmosis in caves	4/8 (50%)	Serology 0/4 (0%) Sputum cytology, culture 4/4 (100%)	2/4 (50%, prolonged symptoms)

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
