# Peer review of "Acute Pulmonary Histoplasmosis Outbreak in A Documentary Film Crew Travelling from Guatemala to Australia"

_tropicalmed, 2019, doi:10.3390/tropicalmed4010025_

Round 1
Reviewer 1 Report
The authors describe an outbreak of histoplasmosis in a filming crew after exposure at a bat cave in Guatemala. five out of the 12 crew members (42%) developed a febrile illness within 1-2 weeks of exposure; 2 were confirmed and 3 were suspected with acute pulmonary histoplasmosis. The exposure, timeline, and presentations fit well with the diagnosis.
lines 59-68: The authors are correct regarding the utility of Histoplasma antigen for the diagnosis of acute pulmonary histoplasmosis, especially early in the presentation (shortly following exposure) where complement fixation and immunodiffusion are not as sensitive. the authors should also mention the utility of the newer serologic test for histoplasmosis, namely the MVD-EIA IgG and IgM test. this antibody test was studied in the setting of outbreak investigation (2 of which are referenced in this manuscript; ref 3,4). please add the following reference to this report and add a statement regarding the utility of EIA Ig testing for acute histoplasmosis. Richer et al, Improved Diagnosis of Acute Pulmonary Histoplasmosis by Combining Antigen and Antibody detection, Clin Infect Dis. 2016 Apr 1; 62(7): 896–902.
Author Response
Dear reviewer,
Thank you for your valuable feedback. I have made a comment and included the reference, as suggested.
Many thanks again,
Steve
Reviewer 2 Report
The case report "Acute pulmonary histoplasmosis outbreak in a documentary film crew travelling from Guatemala to Australia" shows the difficulty of diagnosing histoplasmosis quickly mostly due to the limited availability of diagnostic tests or the long necessary incubation time of cultures. This highlights the importance of an exact patient history which is especally illustrated in this case report.
The report is of clinical relevance, comprehensvie and very well written. I recommend the pubication of this report.
Two minor remarks:
line 17 - did the counselling include patient education regarding histoplasmosis (and the respective symptoms)? Were the masks specifically inteded to prevent histoplasmosis or were they just overall safety measures?
line 68 - I would suggest to highlight that seroloy should be repeated after several weeks if negative on first testing (as done in patient 1) and urine antigen is not available.
Author Response
Dear reviewer,
Thank you for the valuable feedback. I have made the changes as suggested.
To comment on your first two points, the patients just received general pre-travel advice, but not specific to histoplasmosis. I believe these cases are illustrative of the need for such advice! The patients were provided with masks as an overall safety measure (but not reinforced, as such).
I have also made the changes as suggested in your second point.
Many thanks again,
Steve
Reviewer 3 Report
This is a case report describing details of two of five returned travellers with acute histoplasmosis acquired in Central America along with a brief literature review summarising other cases. This is a well-written, interesting case report.
I recommend slightly broader consideration of the sensitivity of urinary antigen for diagnosis of acute histoplasmosis in immunocompetent hosts. Although this may be the most sensitive test acutely, it has poor overall performance.
Author Response
Dear reviewer,
Thank you for your valuable feedback. I have made an additional comment, as suggested.
Thanks again,
Steve